# 2′O-Ribose Methylation of Ribosomal RNAs: Natural Diversity in Living Organisms, Biological Processes, and Diseases

**DOI:** 10.3390/cells10081948

**Published:** 2021-07-31

**Authors:** Mariam Jaafar, Hermes Paraqindes, Mathieu Gabut, Jean-Jacques Diaz, Virginie Marcel, Sébastien Durand

**Affiliations:** Inserm U1052, CNRS UMR5286, Centre de Recherche en Cancérologie de Lyon, Université de Lyon, Université Claude Bernard Lyon 1, Centre Léon Bérard, CEDEX 08, F-69373 Lyon, France; Mariam.JAAFAR@lyon.unicancer.fr (M.J.); Hermes.PARAQINDES@lyon.unicancer.fr (H.P.); Mathieu.GABUT@lyon.unicancer.fr (M.G.); JeanJacques.DIAZ@lyon.unicancer.fr (J.-J.D.)

**Keywords:** 2′-O-methylation, rRNA modification, ribosome, ribosome heterogeneity, RiboMeth-seq

## Abstract

Recent findings suggest that ribosomes, the translational machineries, can display a distinct composition depending on physio-pathological contexts. Thanks to outstanding technological breakthroughs, many studies have reported that variations of rRNA modifications, and more particularly the most abundant rRNA chemical modification, the rRNA 2′O-ribose methylation (2′Ome), intrinsically occur in many organisms. In the last 5 years, accumulating reports have illustrated that rRNA 2′Ome varies in human cell lines but also in living organisms (yeast, plant, zebrafish, mouse, human) during development and diseases. These rRNA 2′Ome variations occur either within a single cell line, organ, or patient’s sample (i.e., intra-variability) or between at least two biological conditions (i.e., inter-variability). Thus, the ribosomes can tolerate the absence of 2′Ome at some specific positions. These observations question whether variations in rRNA 2′Ome could provide ribosomes with particular translational regulatory activities and functional specializations. Here, we compile recent studies supporting the heterogeneity of ribosome composition at rRNA 2′Ome level and provide an overview of the natural diversity in rRNA 2′Ome that has been reported up to now throughout the kingdom of life. Moreover, we discuss the little evidence that suggests that variations of rRNA 2′Ome can effectively impact the ribosome activity and contribute to the etiology of some human diseases.

## 1. Introduction

First observed in the mid-1950s, the ribosome is the cellular machinery that synthesizes proteins using mRNAs during the process of translation. The understanding of ribosome activities and functions has evolved with the constant improvement of structural methods [1,2]. The recently-met atomic resolution level of the ribosome’s structure demonstrated that its function has been conserved throughout evolution, from bacteria to higher vertebrates, due to the high conservation of its structure, which is based on two main components: the ribosomal proteins and the ribosomal RNAs (rRNAs) [2]. In humans, ribosomes are constructed of 80 ribosomal proteins organized around 4 rRNAs: 5S, 5.8S, 18S, and 28S [2,3,4]. Although the number and length of ribosomal proteins and/or rRNAs vary across species, the ribosome core displays a similar structuration on which its functions of mRNA-decoding and peptidyl-transferase activities depend [2].

The conserved structuration of rRNAs relies on ribosomal proteins but also to some extent on the presence of chemical modifications at some particular rRNA nucleotides that include 2′O-ribose methylation (2′Ome, Figure 1A), pseudo-uridylation (ψ) and base methylations and acetylations [5,6]. These chemical modifications maintain the modified base in a particular orientation, stabilize RNA:RNA interactions or, in contrast, inhibit RNA:protein association [7,8]. Therefore, based on the strong conservation of the structure/function relationship of the ribosome, the long-lasting dogma was that rRNAs are always and similarly chemically modified and that these modifications are critical for ribosome functions. Accordingly, this vision was supported by observations using non-quantitative methods that, in the two historical models *Saccharomyces cerevisiae* and *Xenopus laevis*, rRNAs are fully methylated at almost all specific positions in all ribosomes [6,9]. Similarly, knocking out the Fibrillarin protein (i.e., the rRNA 2′O-methyltransferase, FBL, Figure 1B) induces lethality in yeast, zebrafish, and mouse models, supporting an essential role of the rRNA 2′Ome across species [10,11]. Therefore, it was broadly admitted that rRNA, and more generally the ribosome, always displays the same composition to sustain its conserved structure and thus functions, and that a disruption of this structure is detrimental for the ribosome activity in living organisms.

However, recently, several data has suggested that ribosomes can actually display distinct and variable compositions, both at ribosomal protein and RNA levels [1,7,12,13,14,15,16]. The current hypothesis proposes that these changes in ribosome composition may directly affect its intrinsic translational properties and thus provide the ribosome with translational regulatory capacities. Thus, rather than simply being a machinery that synthesizes proteins, the ribosome emerges as a direct regulator of translation. However, only some data has supported this hypothesis up to now and efforts must continue to definitively universalize this model. In contrast, we have reached a moment where technological breakthroughs provide strong and numerous evidence that variations in chemical rRNA modifications occur naturally in living organisms, biological processes, and diseases. Indeed, the scientific community has yet accumulated numerous pieces of data in less than five years, demonstrating that an rRNA molecule can subsist even in the absence of chemical modifications at some particular positions, most of these studies focusing on 2′Ome. Here, we compiled and discussed studies providing experimental observations that definitively demonstrate that rRNA 2′Ome varies naturally in living organisms, specifically depending on the physio-pathological context, however.

## 2. A Brief Overview of rRNA 2′Ome

2′Ome consists in the addition of a methyl group (CH3) onto the hydroxyl (OH) of the second carbon of the ribose using the S-adenosyl-methionine (SAM) as a methyl donor (Figure 1A). Out of the 4 rRNAs, modified nucleotides have been reported on the 3 rRNAs of that derived from the common 47S precursor (i.e., 5.8S, 18S and 28S) [6]. The addition of 2′Ome has been shown to occur co-transcriptionally and to participate in the precise shaping of the rRNA structure on which the ribosome activity relies [17,18]. This observation pushed the scientific community to consider that the rRNA 2′Ome is required for building and maintaining ribosome activities.

2′Ome is catalyzed by a methylation complex, the composition of which is conserved in archea and eukaryotes (Figure 1B). This complex contains four different proteins and one non-coding small nucleolar RNA (snoRNA) and thus forms particles called small nucleolar ribonucleoproteins (snoRNPs) [6,7,19]. The addition of 2′Ome occurs at some particular nucleotides along the rRNAs and is guided by the family of snoRNAs that exhibits C/D boxes, called SNORDs [7,20]. Indeed, SNORDs directly interact with rRNAs by sequence homology to guide the rRNA 2′O-methyltransferase FBL onto the nucleotide to be modified. Interactions between SNORDs and rRNAs depend on the particular structure of the methylation complex provided by NOP58, NOP56, and NHP2L1 (also known as 15.5kDa protein) in humans. Among archea and eukaryotes, some 2′Ome sites do not depend on the snoRNP methylation complex but instead on individual site-specific rRNA methyl-transferases [21]. For example, the yeast 25S-Gm2922 and 25S-Um2921 are catalyzed by the stand-alone methyl-transferase Spb1, independently of C/D box snoRNAs [22,23]. Such site-specific rRNA methyl-transferases and their associated 2′Ome sites are conserved from bacteria, where only four 2′Ome sites have been described as 2′O-methylated, the 16S-Cm1402, 23S-Gm2251, 23S-Cm2498, and the highly conserved 23S-Um2552, which are catalyzed by the stand-alone rRNA methyl-transferases RsmI (YraL), RlmB (YfjH), RlmM (YgdE), and RlmE (RrmJ or Ftsj), respectively [24]. Thus, one could easily hypothesize that snoRNA-dependent methylation most likely overcame the complexification of the number of 2′Ome through evolution. Indeed, historical models reported increasing numbers of 2′Ome rRNA positions, from 3 in bacteria, 55 in yeast, and more than 106 sites in humans, which also correlates with the increased length of rRNAs (www-snoRNA-biotoul.fr, accessed on 29 July 2021). Moreover, most of the 2′Ome sites specifically found in eukaryotes concentrate in ribosome functional domains, which are not 2′O-methylated in bacteria, despite being strongly conserved [6]. This observation supports the notion that the presence or absence of 2′Ome at some particular sites may occur without affecting the conserved function of the ribosome.

Similar to the structure of the ribosome, the mapping of the 2′Ome positions across species evolves concomitantly with the improvement of technologies [25,26]. Historically, the 2′Ome detection relied on biochemical and molecular approaches, including, for example, mass spectrometry, primer extension (i.e., detection of reverse transcription pauses using radiolabeled primers) or RTL-PCR (Reverse Transcription Low dNTPs–PCR), a brief description of which is provided in Table 1. The main limitation of these methods remains that several independent experiments are required to fully map the total > 7kb of rRNAs, thus requiring a huge amount of biological materials and labor. The recent development of RNA-seq-based approaches to map rRNA 2′Ome has opened up several avenues in the field [17,27] (briefly described in Table 2; for review and comparison of these methods, see [25]). From 2016, these methods generated an accumulation of data for all rRNA nucleotides in a single experiment using a limited amount of biological material. These novel approaches not only allow the mapping of 2′Ome in novel living organisms, and thus the study of the conservation of rRNA 2′Ome localization through evolution, but also permit researchers for the first time to quantitatively compare rRNA 2′Ome levels at all sites, both in a single cellular population, organ, or tumor (i.e., intra-variability) and between different conditions (i.e., inter-variability).

## 3. Natural Variability of rRNA 2′Ome in Yeast

To date, biochemical mapping and sequencing-based profiling approaches have identified 55 2′Ome sites in *Saccharomyces cerevisiae* rRNAs: 18 sites in the 18S rRNA, 37 sites in the 25S rRNA and no sites in neither the 5S nor the 5.8S rRNAs [6,17,27,33,34]. Several pieces of evidence have supported the notion that rRNA 2′Ome is critical for ribosome biogenesis in yeast. First, the recent comparison of the 2′Ome patterns in chromatin-associated (i.e., nascent transcripts) and mature rRNAs showed that the majority of the 2′Ome occurring on the 18S rRNA was co-transcriptional, while the 2′Ome on the 25S rRNA seemed to occur both co- and post-transcriptionally [17], in agreement with previous results obtained by kinetic labeling experiments [18]. Therefore, rRNA 2′Ome appears absolutely crucial for the early first steps of pre-rRNA folding and the assembly of ribosomal particles. Second, similar to other eukaryotes, the catalysis of 2′Ome in *S. cerevisiae* is driven by the methyl-transferase Nop1, the yeast homologue of vertebrate FBL, which is highly conserved among all species [35]. Conditional depletions of Nop1 alleles are lethal and result in both impairments of pre-rRNA 2′Ome and processing over long periods of times [36]. Similarly, a Nop1 mutant unable to bind the SAM, and thus abolishing its methyl-transferase activity, severely impaired growth, supporting the importance of 2′Ome in yeast [35]. Altogether, these data support the essential role of rRNA 2′Ome in bona fide ribosome biogenesis and yeast growth.

Interestingly, rRNA modifications, including 2′Ome, are not randomly scattered across the ribosome. The 2D and 3D structures of the 18S and 25S rRNAs revealed that the majority of chemically modified nucleotides conserved between different yeast species, such as *S. pombe* and *S. cerevisiae*, cluster in functionally important domains of the ribosome [30,33]. Accordingly, these conserved modified nucleotides are mainly located within the peptidyl-transferase center (i.e., the formation of the peptidyl-transferase bound between two amino-acids), the decoding center (i.e., the recognition between mRNA codon:tRNA anti-codon) and the inter-subunit bridges (i.e., the interaction between the small and the large subunits), suggesting that their functional importance is fundamental and universal. It was therefore predicted that individual depletions of any rRNA chemical modifications could highly impact both ribosome biogenesis and ribosome translational activity. Surprisingly, however, the loss of individual modifications in yeast only impacted slightly, or not at all, with cell viability or ribosome functions [37,38]. Chemical modifications instead act in a cumulative manner, as significant phenotypes were only observed upon the loss of multiple modifications. For instance, the helix 69 (H69) from the inter-subunit bridge, which interacts with both A and P sites, contains five modifications in yeast that are all conserved in humans: one 2′Ome (25S-Am2256) and four pseudo-uridines (25S-Ψ2258, 25S-Ψ2260, 25S-Ψ2266 and 25S-Ψ2264). The loss of one or two chemical modifications through the depletion of corresponding snoRNAs results in little or no effect, whereas the combinatorial loss of three to five modifications has many effects, including an impairment of cell growth, an increased translation rate and termination at stop codons, a reduced accumulation of rRNA, as well as alterations of ribosome structure and an increase in sensitivity to antibiotics [37]. In addition, the loss of three modifications (18S-Gm1428, 18S-Cm1639, and 18S-m1acp3Ψ1191) in the decoding center reduced growth rates and protein synthesis efficiency and caused a strong deficiency in small subunit production [38]. Although these studies do not address phenotypes caused by only 2′Ome depletions, it nonetheless demonstrates that different combinations of rRNA chemical modification depletions affects the ribosome to various extents, however without being systematically lethal. Hence, unlike initially thought, yeast ribosomes can handle partial losses of chemical modifications.

Importantly, although it was originally thought that individual 2′Ome were of little impact on cell growth and translation, this idea was later challenged in yeast by high-resolution phenotyping experiments. This approach indeed revealed that individual snoRNA depletions could result in important underlying deregulations under stress conditions [39]. Systematic individual deletions of 20 SNORDs result, in most cases, in the alteration of growth adaptation, while the strain’s growth rate is slightly or not at all affected. In addition, the majority of these SNORD deletions result in resistance to several antibiotics including neomycin, paromomycin, and anisomycin, while others result in antibiotic-sensitivity [39]. Interestingly, differences in phenotypic impacts on growth adaptation were not correlated to the spatial distribution of 2′Ome, therefore suggesting that each single 2′Ome fulfills a specific function during rRNA structuration. These data also suggest that the loss of individual rRNA 2′Ome is not strictly critical for ribosomes, although it can modulate ribosome functions in translational regulations in response to environmental changes.

Interestingly, such data not only challenged the long-standing assumption that 2′Ome sites must be constitutively present on specific rRNA residues, but also fed the arising concept that ribosomal heterogeneity naturally occurs. Intriguingly, the level of 2′Ome at each individual position in yeast rRNAs is not the same, and a subset of 2′Ome positions was found to be partially methylated [17,33]. This implies that, in a single yeast population, a mix of non-methylated and fully methylated rRNA molecules at a given position co-exists, corresponding to the notion of rRNA 2′Ome intra-variability. The first study to demonstrate a ribosome heterogeneity at the rRNA 2′Ome level in *S. cerevisiae* used RNA-cleaving DNAzymes coupled to LC-UV MS/MS analysis to show that the methyl group at the ribose of nucleotide A at position 100 of the 18S rRNA (18S-Am100) was missing in one-third of 18S rRNA molecules, therefore revealing an intra-variability of this residue in a single yeast strain [40]. Notably, ribosomes devoid of this 2′Ome still participated in translation, but the exact function of these particular ribosomes was not yet elucidated [40]. Later on, many other studies using different high throughput techniques, including high resolution RP-HPLC analysis, SILNAS-based MS-technology and RiboMeth-seq, revealed that 18S-Am100 was not the only site to be partially methylated in yeast. In fact, while for a given ribosome population the majority of the sites are fully methylated with a methylation score > 0.95, other positions are only partially methylated and display a methylation score < 0.8–0.9 [17,27,33,34]. Interestingly, partially methylated nucleotides are located at similar positions in both *S. cerevisiae* and *S. pombe* rRNAs, therefore indicating that 2′Ome intra-variability displays some degree of conservation [33].

Biological explanations on how and why intra-variability takes place in yeast still remains to be elucidated. We can, however, easily hypothesize that rRNA 2′Ome variability may be adjusted by the regulation of the methylation complex assembly and/or association to targeted pre-rRNA under different contexts. In particular, the modulation of the pre-rRNA/SNORD association is a crucial event during ribosome biogenesis, that is usually regulated by SNORD expression and processing or the catalytic activity of RNA helicases [41,42]. Along these lines, it was shown that Ppr43 and Dbp3, two RNA helicases implicated in the release of several snoRNAs from pre-rRNAs, may modulate rRNA 2′Ome through the regulation of snoRNAs dynamics [43]. Aquino and colleagues showed that Dbp3 depletion in yeast reduced the level of 2′Ome at 18 sites of the 25S rRNA and at 1 site of the 18S rRNA, most likely by trapping snoRNAs on pre-rRNA and therefore interfering with the access of adjacent snoRNPs to mutually exclusive overlapping regions required for the proper completion of 2′Ome. Similarly, Prp43 depletion led to the reduction of 2′Ome level at 8 sites in the 18S rRNA and 25 in the 25S rRNA by a mechanism similar to Dbp3, albeit regulating a specific subset of SNORDs. Although this experimentally driven study constitutes nonetheless strong evidence of 2′Ome modulations by RNA helicases, additional mechanisms that regulate the natural intra-variability in rRNA 2′Ome observed in yeast remain to be explored.

## 4. Variation of rRNA in Metazoan Cell Lines

In addition to yeast models, the variability in rRNA 2′Ome level has also been reported in metazoan cell lines, mostly in human cells and to a lower extent in mouse cells. Hence, in addition to supporting the notion of intra-variability of rRNA 2′Ome within a single cell population as observed in yeast, ex vivo metazoan models also raised the notion that rRNA 2′Ome varies depending on the physio-pathological context, including cell lines harboring clinically relevant specific disease-related gene mutations (Figure 2).

### 4.1. Intra-Variability of rRNA 2′Ome in Human Cell Lines

The existence of variations in rRNA 2′Ome in humans first emerges from analyses of individual human cell lines. Albeit supported by a relatively low number of studies, evidence for the existence of partially methylated rRNA within a single human cell line has started to emerge in 2016 with the description of RNA-seq based approaches [17,26]. Using the RiboMeth-seq approach, which relies on the property of the 2′Ome to protect the methylated nucleotide from alkaline hydrolysis, it was first demonstrated that about one-third of the 106 sites are partially methylated within a single human cell line [26,44,45]. Indeed, the comparison of rRNA 2′Ome levels within a cell population issued from either the cervix adenocarcinoma HeLa cell line or the colorectal carcinoma HCT116 cell line indicated that most of the rRNA 2′Ome positions are fully methylated and that only a subset of rRNA 2′Ome positions are partially methylated. Therefore, the existence of partially methylated sites at particular positions implies a mix of fully methylated and non-methylated molecules in the bulk of analyzed human rRNAs. Furthermore, another study using the high-throughput RibOxi-seq approach confirmed the presence of partially methylated sites in the human ovary teratoma-derived PA1 cell lines [32].

While the next-generation sequencing methods are starting to be widely used, especially RiboMeth-seq, Taoka and colleagues used a quantitative mass spectrometry approach to quantify the level of rRNA modifications in the TK6 cell line (human lymphoblast from spleen) [46]. In addition to quantifying methylation at the 106 known sites, they also identified and quantified six additional rRNA 2′Ome sites (18S-Um354, 18S-Cm621, 18S-Um1668, 28S-Am1310, 28S-Um1760, 28S-Gm3606). Among all of these, about one-third (*n* = 40) was partially methylated and displayed a methylation ratio < 90%, therefore cohering with the proportion of partially methylated sites observed in HeLa cells using RiboMeth-seq [26,44,46]. Interestingly, high-resolution structures of HeLa ribosomes obtained by cryo-EM also identified a subset of rRNA sites partially methylated, although all the sites have not been analyzed by cryo-EM due to heterogeneity in high-resolution within the ribosome structure [1,5].

Altogether, these studies demonstrated that, as in yeast, the presence of 2′Ome at some particular rRNA positions is not compulsory in human cell lines. Cells can thus tolerate the production of ribosome with an absence of 2′Ome, however not at all of the rRNA 2′Ome positions, but only at about one-third of them. However, the distinct sensitivity and scoring of the diverse technical approaches prevent a direct comparison of the intra-variability in these different human cell lines. Moreover, analyses of cell lines in bulk do not allow us to establish whether the 2′Ome intra-variability results from intra-cellular 2′Ome variations of rRNA molecules or, in contrast, reflects a cell-to-cell heterogeneity of 2′Ome. Nevertheless, these data indicate that ribosomes of distinct 2′Ome compositions co-exist in human cell lines, supporting the notion of ribosome heterogeneity.

### 4.2. Physiopathological Variability of rRNA 2′Ome in Cell Lines

In parallel to the increasing evidence for natural intra-variability of 2′Ome in cell lines, several studies have pointed out that artificial and experimental manipulations of genes, involved in ribosome biogenesis or physio-pathological processes, may induce variations of 2′Ome between two different biological conditions, supporting the notion of inter-variability in rRNA 2′Ome (Figure 2).

#### 4.2.1. rRNA 2′Ome Variability in Response to Modulations of Ribosome Biogenesis Factor (RBF) Expression and Activity

Some studies first examined the impact of RBFs on rRNA 2′Ome regulations, such as the recent work that Wu and colleagues dedicated to the long non-coding RNA *ZFAS1*, a host gene that encodes three SNORD12 family members (SNORD12, SNORD12B, and SNORD12C) [47]. In a panel of human colorectal cancer cell lines, it appeared that the depletion of *ZFAS1* decreases the level of rRNA 2′Ome at 28S-Gm3878 and 28S-Gm4593, guided by SNORD12C and SNORD78, respectively. Indeed, in addition to encoding SNORDs, ZFAS promotes the assembly of SNORD12C- but also SNORD78-containing snoRNP complexes, through a direct interaction with a member of the methylation complex, NOP58. Although the ZFAS1 knockdown decreases cell proliferation and invasion of colorectal cancer in vitro and in vivo, the direct implication of SNORDs and rRNA 2′Ome has not been demonstrated. In addition, the impacts of other snoRNAs, the expression of which is deregulated in cancer, on variations of rRNA 2′Ome has also been reported. For instance, SNORA23, which is an H/ACA box snoRNA downregulated in hepatocellular carcinoma tissues and cell lines, surprisingly functions by impairing the 2′Ome at 28S-Cm4506 in Huh7 cell lines [48]. In this study, SNORA23 expression was shown to be regulated by the PI3K/AKT/mTOR signaling pathway and the increased SNORA23 expression inhibits proliferation, migration, and invasion, although the contribution of 28S-Cm4506 has not been addressed.

Along the lines of the ZFAS study, many other studies have demonstrated that modulations of methylation complex expression and formation induced variations of rRNA 2′Ome in cell lines. For instance, *FBL* depletions in HeLa cells caused a global decrease in rRNA 2′Ome levels at most, albeit not all, positions [44,49]. As a matter of fact, RiboMeth-seq analyses showed that the decrease in rRNA 2′Ome was surprisingly site-specific, rather than systematic. In addition, partially methylated sites exhibited a peculiar sensitivity to *FBL* depletion, as sites with a methylation score < 0.8 in control HeLa cells decreased by at least 10% upon *FBL* depletions. Importantly, functionally critical regions of the ribosome, including the peptidyl-transferase and decoding centers, which contain the most conserved modification sites, were devoid of 2′Ome alterations upon *FBL* depletions, in contrast to the inter-subunit bridges, peptide exit tunnel, and A and P-sites [44]. Notably, the decrease in rRNA 2′Ome at particular sites upon *FBL* depletions was neither the result of a reduction in rRNA production nor a consequence of changes in corresponding snoRNA expression [44,49]. These data suggest that the full methylation at specific sites seems to be required for ribosome structure and functions, while partial methylation of others can be tolerated. Thus, one can hypothesize that modulations of FBL activity and expression could be a means to finely tune ribosome functions by modulating the 2′Ome of specific sites without affecting the full methylation at key positions. Another study has recently highlighted the importance of the methylation complex formation in the regulation of rRNA 2′Ome [50]. EZH2, a lysine methyltransferase catalyzing histone H3 lysine 27 trimethylation (H3K27me3) and member of the polycomb repressive complex 2 (PRC2) was found to interact with components of the rRNA methylation complex and thus play a role in 2′Ome independently from its chromatin-related functions through histone methylation [50]. More precisely, it was shown that EZH2 binds strongly and directly to both the methylation complex proteins FBL and NOP56. EZH2 is required for the assembly of the methylation complex as its depletion impairs the assembly of core proteins NOP56, NOP58, and SNU13. Accordingly, EZH2 depletions reduce site-specific 2′Ome in prostate C4-2 cancer cells compared to control cells, with 2 sites affected in the 5.8S rRNA, 31 in the 18S rRNA, and 54 in the 28S rRNA. Importantly, the *FBL* overexpression was found to rescue the decrease in 2′Ome caused by EZH2 depletions, therefore suggesting that EZH2 modulates 2′Ome through FBL methyl-transferase activity. Finally, EZH2 promotes the Internal Ribosome Entry Site (IRES)-dependent translation of several genes, including XIAP, although the implication of FBL/NOP56 or rRNA 2′Ome in this process remains to be proven [50].

#### 4.2.2. rRNA 2′Ome Variability in Physio-Pathological Contexts

The question of an inter-variability of 2′Ome occurring in physio-pathological contexts as well as its contribution to disease’s etiology has recently gained much attention. Indeed, by using human cell lines harboring clinically relevant genetic mutations, some recent studies have provided evidence that a 2′Ome variability occurs in specific pathologies, although only few of them provide convincing data supporting the key contributions of 2′Ome to disease’s phenotypes (Figure 2).

The fragile X syndrome results from the loss-of-function of the RNA-binding Fragile X Mental Retardation Protein (FMRP), which has been recently connected to the regulation of rRNA 2′Ome in human Embryonic Stem cells (hESC) [51]. D’Souza and colleagues observed that FMRP interacts with several SNORDs to form FBL-independent snoRNPs. Importantly, RiboMeth-seq analyses showed that *FMRP* knock-out in hESC significantly affected 2′Ome at 17 positions, including 18S-Cm174, 18S-Um428, 18S-Gm867, 18S-Um1248, 28S-Um2402, 28S-Cm2409, 28S-Um4276, and 28S-Cm4917 (nomenclature from original article corrected). As reported with FBL, *FMRP* depletions mostly resulted in alterations (although both up and down) of partially methylated sites, while fully methylated ones remained unaffected [44,49,51]. Since levels of SNORDs guiding methylations at these positions were not affected, further studies will be required to get more of an insight into the mechanisms by which FMRP either promotes or represses specific rRNA 2′Ome. Moreover, the authors did not investigate whether patients with FMRP-dependent fragile X syndrome display 2′Ome alterations and whether these potential alterations contribute to disease phenotypes.

Another outstanding illustration of the implication of rRNA 2′Ome variations in human genetic disorders has been provided by a recent work focused on the ribosome biogenesis factor nucleophosmin 1 (NPM1) [52]. Nachmani and colleagues observed that artificial depletions of *NPM1* in mouse cell lines altered 2′Ome at positions 28S-Cm1327, 28S-Am3764, 28S-Cm3866, 28S-Um3904, and 28S-Gm4198 (human nomenclature). In addition, the human leukemia OCI-AML3 cell line, which expresses a mutated form of NPM1 sequestered in the cytoplasm (NPMc^+^) and frequently found in Acute Myeloid Leukemia (AML) [53], harbors reduced 2′Ome at positions 28S-Cm1327, 28S-Cm3866, 28S-Um3904, and 28S-Gm4198, while respective SNORD levels remains unchanged. In the same study, Nachmani and colleagues also identified two NPM1 mutations, NPM1^D178H^ and NPM1^D180del^, in patients with Dyskeratosis Congenita (DC), where causative mutations could not have been identified. These mutations reduced NPM1 abilities to interact with snoRNAs (SNORD15/47/52/58/104) and decreased the 2′Ome at corresponding positions (28S-Am3764, 28S-Cm3866, 28S-Um3904, 28S-Gm4198, and 28S-Cm1327). Additionally, a mouse model carrying the NPM1^D180del^ mutation recapitulates DC pathological features. Interestingly, NPM1 depletions, cytoplasmic retention of NPM1 (NPMc^+^), and NPM1^D178H^/NPM1^D180del^ mutations impacted the translation of specific IRES-containing transcripts. Altogether, the study by Nachmani et al. provided solid evidence that a variability of 2′Ome at specific positions exists in certain forms of NPM1-dependent DC and leukemia, and most likely participates in the physiopathology of these diseases. However, further investigations should carefully address the contribution of rRNA 2′Ome-dependent translational modulations to the etiology of the diseases.

Several other studies have highlighted the existence of a 2′Ome variability in other forms of AML and have demonstrated its contribution to leukemogenesis in a NPM1-independent manner [54,55]. Hence, the oncogenic AML1-ETO fusion protein, which results from the most common balanced translocation in AML (t(8:21)), plays essential roles in hematopoietic progenitor cell (HPC) self-renewal, in part by promoting the expression of groucho-related Amino-terminal Enhancer of Splicing (AES) [56,57,58]. Strikingly, *AES* knock-down in the human Kasumi-1 cell line impacted the abundance of many SNORDs and, accordingly, *AES* depletion tremendously reduced 2′Ome at three particular positions (18S-Cm1703, 18S-Gm1328, and 28S-Cm4506) while pseudo-uridylation was only weakly affected [55]. Notably, the authors did not perform high-throughput analyses of rRNA 2′Ome but most likely focused on positions guided by candidate SNORDs, the level of which was affected by *AES* depletion. Mechanistically speaking, the authors showed that AES interacts with the RNA helicase DDX21 and is required for DDX21 interactions with proteins involved in the methylation complex, including FBL, NOP56, and NOP58, as well as in pre-rRNA and rRNA processing complexes. Interestingly, AML1-ETO positive patients’ primary AML blasts overexpress specific SNORDs, including SNORD14D/14E/20/32A/34/35A/43/53/104 and displays increased 2′Ome at positions 18S-Um1805, 28S-Um4197, 28S-Cm3848, 18S-Cm1703, and 18S-Gm1328. Furthermore, a set of snoRNAs, including SNORD20/34/35A/43, were found to be associated with AML displaying a high leukemic stem cell content and a strong treatment resistance. Altogether, this remarkable work by Zhou et al. demonstrated that increased snoRNA expression, and therefore 2′Ome levels at specific rRNA positions, occurs during AML1-ETO-dependent leukemogenesis and contributes to both the physiopathology and the severity of the disease. In a follow-up snoRNA screening study, the team of Müller-Tidow also identified SNORD42A and 18S-Um117 as major actors of leukemogenesis in AML cell lines and patient’s blast [54]. Altogether, these studies shed light on the role of specific 2′Ome in leukemogenesis and possibly normal hematologic functions. On one hand, rRNA 2′Ome are required for normal hematopoiesis and bone marrow functions, and the variability of 2′Ome at these positions is observed in NPM1-related leukemia or DC and contributes to the disease’s phenotypes [52]. On the other hand, high expressions of specific SNORDs and elevated 2′Ome levels at related specific positions are observed in AML1-ETO-, and possibly C-MYC- and MLL-AF9-associated AML, and participate in leukemogenesis [54,55]. Of note, while the FMRP- and NPM1-mediated 2′Ome variabilities imply changes in methylation complex formations without affecting the SNORD expression, AML1-ETO alternatively controls specific 2′Ome changes by regulating the expression of specific SNORDs.

Additional key oncogenes and tumor suppressors have been associated with alterations of rRNA 2′Ome. A preprint study available on Research Square focused on the MYC proto-oncogene [59]. BJ^hTERT^ fibroblasts expressing MYC in an inducible manner showed that MYC overexpression is associated with an increase in *FBL* expression and changes in 2′Ome levels at three sites: a decrease at 28S-Um2031 and 28S-Um2402, and an increase at 18S-Cm174 [59]. Interestingly, SNORD45C, which targets 18S-Cm174, was also found up-regulated under MYC overexpression. *SNORD45C* knock-out and the related abrogation of 2′Ome at 18S-Cm174 in HeLa cells resulted in translational deregulations of a large number of specific mRNAs involved in cell cycle, mitosis, metabolism, and intracellular transport, for example [59]. Interestingly, mRNAs up- and down-regulated upon SNORD45C depletion displayed a strong variation in elongation-related codon compositions, suggesting that translational defects caused by SNORD45C depletion might depend on ORF sequences and codon usage. However, it is unclear whether or not the effects observed by the authors were directly dependent on perturbations of rRNA 2′Ome.

In addition, several studies focused on *TP53*, a tumor suppressor gene that is frequently inactivated in solid tumors mainly by mutations (for reviews [60,61]) and investigated its functions in rRNA 2′Ome. It is now well established that p53 suppresses RNA Pol I transcription and its inactivation in cancer is therefore directly associated with the up-regulation of RNA Polymerase I activity and ribosome production [62]. Thus, it was hypothesized that p53 could also affect rRNA 2′Ome. Using an isogenic colorectal cancer cell line HCT116, Sharma and colleagues reported using RiboMeth-seq, and that, among the 22 partially methylated sites, 13 sites showed a decrease in rRNA 2′Ome level in p53 −/− cell line compared to p53 +/+ one, demonstrating that some 2′Ome sites are particularly sensitive to the absence of functional p53 [49]. Of note, it was reported that *FBL* is a direct target of p53 [63]. In human mammary epithelial cells (HME) and in the HCT116 colorectal cell lines, p53 inactivation increases *FBL* expression and promotes a significant alteration of 2′Ome levels at several specific sites in the 5.8S, 18S and 28S rRNAs, although RTL-qPCR technic failed to draw a full overview of alterations of rRNA 2′Ome. Importantly, the alteration of rRNA 2′Ome was not correlated to expression levels of corresponding SNORDs, suggesting once more that 2′Ome defects are strictly dependent upon SNORDs expression [63]. Importantly, in agreement with 2′Ome responses to NPM1 alterations, p53- and FBL-induced alterations of rRNA 2′Ome impact IRES-dependent translation of some mRNAs, and in vitro translation assays have demonstrated that variations in rRNA 2′Ome are responsible for changes in IRES-dependent translation [44,63]. Altogether, these data demonstrate a direct link between p53, *FBL* and site-specific rRNA 2′Ome regulation, although the reasons behind the differential sensitivity of specific 2′Ome to p53 remains an open question that needs to be addressed. Interestingly, variations in rRNA 2′Ome levels have also been shown to increase at some specific sites in cancer cells displaying enhanced aggressivity [64]. Hence, human breast cancer MCF-derived cells, which exhibit a reduced ARL2 (ADP ribosylation factor like 2) expression associated with enhanced in vivo and in vitro tumor aggressivity, display increased 2′Ome levels at six out of eleven sites analyzed in the 18S and 28S rRNAs compared to wild-type cells. Although this observation needs to be broadened by systematically analyzing all rRNA 2′Ome sites, it is nonetheless a clear indicator of qualitative alterations of ribosome composition in aggressive tumors.

Finally, a few pieces of evidence suggest that cellular stresses could also impact 2′Ome. Hence, Metge and colleagues investigated the effect of hypoxia on ribosome biogenesis in T47D and MCF10 breast cancer cell lines and found that not only does hypoxia up-regulate RNA pol I activity and rRNA synthesis compared to normoxia but also modifies 2′Ome of seven out of fourteen sites measured by RTL-qPCR [65]. Interestingly, ribosomes bound to the IRES located in the 5′UTR of *VEGF-C* mRNA displayed distinct 2′Ome patterns in hypoxia compared to normoxia, therefore suggesting that *VEGF-C* mRNA could be translated by a distinct pool of ribosomes upon changes in oxygen supply, with potential consequences during tumorigenesis. Although the molecular mechanism behind the modulation of 2′Ome during hypoxia remains to be fully elucidated, it seems that a decrease in the expression of NMI (“N-Myc and STAT interactor”) may impede the formation of methylation complexes containing specific snoRNAs. In addition to hypoxia, genotoxic stresses induced by etoposide, a Topoisomerase II inhibitor commonly used as a treatment for multiple cancers, has been shown to promote changes in 2′Ome [66]. Accordingly, etoposide increases levels of SNORD68 and SNORD111B and therefore augmented 2′Ome at 28S-Am2388 and 28S-Gm3923 in HeLa cells. Considering the wild impact of etoposide on cellular processes, the mode of action could be quite complex and multimodal. However, the authors proposed that etoposide could interfere with SMN and Drosha functions in the processing of nucleolus-enriched, small Cajal body-specific (sca)RNAs, which were hypothesized to regulate rRNA methylation complex activity [66,67,68].

Altogether, these data support that intra- and inter-variability of rRNA 2′Ome occurs in physiologically and clinically relevant cellular models (Figure 2). Profiling of rRNA 2′Ome should now be performed in human disease-related samples to definitively demonstrate that rRNA 2′Ome can occur in these different pathological contexts.

## 5. Diversity of rRNA 2′Ome Profiles in Human Cancer Tissues

Until the recent advances in omics approach to analyzing rRNA 2′Ome using RNA-seq-based methodologies, demonstrating the natural variation in rRNA 2′Ome, as well as evaluating its diversity in human tissues, was a real challenge. Indeed, due to the usual small quantity of available human tissue material, only methods related to RTL-P could be performed and therefore only a limited number of sites could be addressed [29,64]. To overcome the material quantity limitation, the earliest studies mainly investigated human samples from hematological diseases. Cell isolations from blood samples indeed allowed purifications of large amounts of total RNA that can be analyzed by RTL-P. Thus, by using human blood samples, two studies reported alterations in rRNA 2′Ome in acute myeloid leukemia (AML), albeit in different contexts of oncogenic alterations [52,55]. A significant increase in rRNA 2′Ome level at two out of the eleven rRNA sites analyzed was observed in five AML patients carrying the fusion AML1-ETO alteration compared to five heathy donors (i.e., 18S-Cm1703 and 18S-Gm1328) [55]. However, only three of the AML samples causes this increase, suggesting that the specific increase in rRNA 2′Ome at 18S-Cm1703 and 18S-Gm1328 is not a universal trait of AML1-ETO AML patients. In contrast to AML1-ETO1-driven AML, a comparison of 16 NPM1c^+^-driven AML patient samples with 14 samples from patients with clinical remission using RTL-P revealed a decrease at the four sites analyzed (i.e., 28S-Cm1327, 28S-Cm3866, 28S-Um3904, and 28S-Gm4198). Interestingly, the two studies found different changes in rRNA 2′Ome at different sites in AML patients compared to healthy donors, suggesting a specificity of 2′Ome variations depending on the nature of the oncogenic activation. At present only one study investigated variation in rRNA 2′Ome in non-cancer-related disease [69]. In a cohort of 15 heterozygous ß-thalassemic patients and 15 healthy donors, a significant increase in the level of rRNA 2′Ome in two out of the three rRNA sites analyzed by RTL-qPCR (i.e., 28S-Am1858 and 28S-Cm4506) was observed, further supporting the fact that specific alterations of rRNA 2′Ome at some particular sites occur in human disease. Although RTL-P-based approaches allowed researchers to gain an insight into the variability of rRNA 2′Ome at some particular sites in human pathological tissues, such approaches failed to draw the complete landscape of rRNA 2′Ome profile diversity in these conditions.

In late 2020, the first evaluation of the complete diversity of rRNA 2′Ome in human cancer samples was established using the RiboMeth-seq technology by two labs in back-to-back publications [70,71]. As in previous RTL-P analyses performed on blood samples, one study established the rRNA 2′Ome profile of 17 diffuse large B-cell lymphoma (DLBCL) patient samples and three healthy donors using an IonTorrent-based RiboMeth-seq approach [70]. In this cohort, the difference between rRNA 2′Ome levels of each individual DLBCL samples and the mean of rRNA 2′Ome levels of normal samples was estimated for 110 rRNA 2′Ome sites. Although some DLBCL samples showed a significant increase in their rRNA 2′Ome at some sites compared to mean normal samples (i.e., 12 sites), most of the significant difference corresponded to a decrease in rRNA 2′Ome at some sites (i.e., 65 sites). Two main hypotheses might explain these variations in rRNA 2′Ome. First, the decrease in SNORD expression may be responsible for the decreased rRNA 2′Ome levels, especially for the three sites with the most important variation, as observed in AML1-ETO-dependent AML [55]. Second, the global decrease in rRNA 2′Ome could be caused by a passive effect driven by the hyperproliferation of cells, as the significant decrease in 2′Ome correlates with an increase in the proliferative Ki67 labelling index [70]. Indeed, rRNA synthesis is hyperactivated in cancer cells to maintain a high level of protein synthesis, essential for sustaining an hyperproliferative rate [72]. In this condition, a decoupling between the hyperproduction of rRNA and the limited expression of factors involved in rRNA 2′Ome, such as SNORDs, might explain a broad decrease in rRNA 2′Ome level. Based on this latter hypothesis, variations in rRNA 2′Ome can be interpreted as a biological noise. However, the observed increase in the rRNA 2′Ome level in some DLCBL and AML1-ETO-driven AML patients, and the correlation between SNORD expression and rRNA 2′Ome levels suggests that the variation of rRNA 2′Ome, at least at some sites, is cancer-specific or even tumor-specific [32,52,70]. In addition to being specific in regard to the nature of the tumor, rRNA 2′Ome levels are also different at some particular sites when comparing three distinct normal human organs (i.e., brain, liver, and skeletal muscle), supporting the tissue-specificity of rRNA 2′Ome [70].

The second study performed the world’s first analysis of rRNA 2′Ome (i) as a global profile rather than in site-by-site, thanks to the Illumina-based RiboMeth-seq approach, (ii) in solid tumors, and (iii) using a large series compiling more than 200 human samples [71]. Thus, the analysis of 195 primary, non-metastatic breast tumors at 106 rRNA 2′Ome sites supports the previous observations obtained in cell lines: within a single human sample, a mix of differentially methylated rRNA molecules was present, with sites being always methylated while others existed either methylated or not. As in previous studies, specific rRNA 2′Ome alterations were observed in tumors compared to healthy tissues. Interestingly, a significant decrease in rRNA 2′Ome levels was identified for one particular site in a second series of 10 malignant mammary tumors and 8 benign tumors (i.e., 18S-Am576) [71]. Thus, this observation supports the notion that some alterations of the rRNA 2′Ome level can correlate with the pathological degree of the sample. Moreover, as observed in DLBCL, the analysis of the 195 breast tumors revealed that rRNA 2′Ome, at some sites, are naturally different from one patient to another, suggesting a tumor-specific variability of rRNA 2′Ome [70]. More importantly, it appears that, while 60% of the rRNA sites displayed a very small variation in their 2′Ome level between the 195 breast cancer patients reflecting the full heterogeneity of breast tumors, about 40% of the rRNA sites exhibited a high variation [71]. The “stable” rRNA 2′Ome sites are conserved throughout evolution and are concentrated directly within main functional domains of the ribosome. In contrast, the “variable” rRNA 2′Ome sites appear during evolution and localize both at the second layer of the main ribosomal functional domains and at positions of direct interaction with ribosomal proteins or core translational factors. This demonstration of the co-existence of compulsory and plastic 2′Ome at specific rRNA positions reconciles two divergent visions of rRNA 2′Ome. First, it unambiguously challenges the long-lasting notion that the presence of rRNA chemical modifications is strictly required for maintaining a proper ribosome structure, which is necessary to support the bona fide ribosome’s intrinsic activity and therefore cell fitness [6,10,11,73]. In contrast, this finding supports the emerging notion that rRNA chemical modifications may provide plasticity to the ribosome structure and therefore provide the ribosome with regulatory abilities [44,52,63].

Finally, the study by Marcel and colleagues reported that differences in rRNA 2′Ome at the “variable” sites was also associated with biological and clinical features [71]. For example, a significant decrease in rRNA 2′Ome levels was observed in triple-negative breast cancers compared to luminal breast cancers at 18S-Gm1447, while a significant increase was observed at 18S-Am576, therefore supporting specific alterations of rRNA 2′Ome associated with the biological and clinical context. In addition, based on rRNA 2′Ome profiles at 106 positions, four groups of breast tumors, which displayed different overall survival and grade repartition and intrinsic breast cancer subtypes, have been identified. Notably, the breast tumor group that displayed a global decrease in rRNA 2′Ome compared to others exhibited the poorest overall survival and the highest tumor grade, usually associated with a hyperproliferative state. Therefore, in the same manner as the transcriptomic approaches, the rRNA 2′Ome signature may recapitulate several oncogenic and biological traits of tumors. To this regard, we may expect to not only identify a high diversity of rRNA 2′Ome profiles in cancer and, more largely, in human diseases, but also to be able to use rRNA 2′Ome profiling as a direct readout of the inter-tumoral heterogeneity. Further studies will have to address the challenging question of whether the intra-tumoral variability of rRNA 2′Ome reflects either an intra-tumoral cell heterogeneity or a cell-to-cell variability in rRNA modifications.

Overall, this first profiling of rRNA 2′Ome in tumor samples demonstrated the high diversity of rRNA 2′Ome in humans and the substantiate occurrence of natural variation of rRNA 2′Ome in real-life. More importantly, these studies emphasize the notion that variation in rRNA 2′Ome, and even more so at particular rRNA 2′Ome sites, is not random but rather correlates with something biological and clinical, therefore unraveling the potential role of the rRNA 2′Ome in biology. 

**Table 2 cells-10-01948-t002:** The twelve most referenced human rRNA 2′Ome sites showing variability.

Site ^1^	Biological Context of Variability	Variation	Related snoRNA (Host Gene)	Method	Reference
18S-Um116	FBL depletion	Down	SNORD42A/B	RMS ^2^-Illumina	[44]
	SNORD42A depletion	Down	(RPL23A)	RMS-Illumina + RTL-P	[54]
	EZH2 depletion	Down		RMS-Illumina	[50]
	Cell lines, between breast cancer patients	Up and down		RMS-Illumina	[71]
18S-Cm174	FMRP depletion	Up	SNORD45C	RMS-Illumina	[51]
	FBL depletion	Down	(RABGGTB)	RMS-Illumina	[44]
	c-MYC overexpression	Up		RMS-Ion Torrent	[59]
	EZH2 depletion	Down		RMS-Illumina	[50]
	Cell lines	Up and down		RMS-Ion Torrent	[26]
18S-Gm867	FMRP depletion	Down	SNORD98	RMS-Illumina	[51]
	EZH2 depletion	Down	(CCAR1)	RMS-Illumina	[50]
	FBL depletion	Down		RMS-Illumina	[44]
	Cell lines, between breast cancer patients	Up and down		RMS-Illumina	[71]
28S-Cm1327	EZH2 depletion	Down	SNORD104	RMS-Illumina	[50]
	NPM1 depletion and loss-of-function, between AML patients	Down	(lncSNHG25)	RTL-P	[52]
	FBL depletion	Down		RMS-Illumina	[44]
	Cell lines, between breast cancer patients	Up and down		RMS-Illumina	[71]
28S-Am1858	Reduced Arl2 expression	Up	SNORD38A/B	RTL-P	[64]
	FBL depletion	Down	(RPS8)	RMS-Illumina	[44]
	EZH2 depletion	Down		RMS-Illumina	[50]
	DLBCL patients vs. healthy donor	Down		RMS-Ion Torrent	[70]
	ß-thalassemic patients vs. healthy donor	Up		RTL-P	[69]
28S-Um2402	FMRP depletion	Down	SNORD143, SNORD144	RMS-Illumina	[51]
	FBL depletion	Down	(SEC31A)	RMS-Illumina	[44]
	c-MYC overexpression	Down		RMS-Ion Torrent	[59]
	cell lines	Up and down		RMS-Ion Torrent	[26]
	Cell lines, between breast cancer patients	Up and down		RMS-Illumina	[71]
	DLBCL patients vs. healthy donor	Down		RMS-Ion Torrent	[70]
28S-Cm2409	EZH2 depletion	Down	SNORD5	RMS-Illumina	[50]
	FMRP depletion	Up	(TAF1D)	RMS-Illumina	[51]
	FBL depletion	Down		RMS-Illumina	[44]
	cell lines	Up and down		RMS-Ion Torrent	[26]
	cell lines, between breast cancer patients	Up and down		RMS-Illumina	[71]
28S-Am2774	FMRP depletion	Down	SNORD99	RMS-Illumina	[51]
	FBL depletion	Down	(SNHG12)	RMS-Illumina	[44]
	Cell lines, between breast cancer patients	Up and down		RMS-Illumina	[71]
	Brain, liver and skeletal muscle	Down in Liver		RMS-Ion Torrent	[70]
28S-Cm2848	EZH2 depletion	Down	SNORD50A/B	RMS-Illumina	[50]
	FBL depletion	Down	(SNHG5)	RMS-Illumina	[44]
	Cell lines	Up and down		RMS-Ion Torrent	[26]
	Cell lines, between breast cancer patients	Up and down		RMS-Illumina	[71]
28S-Gm3923	FMRP depletion	Down	SNORD111/B	RMS-Illumina	[51]
	FBL depletion	Down	(SF3B3)	RMS-Illumina	[44]
	Cell lines	Up and down		RMS-Ion Torrent	[26]
	cell lines, between breast cancer patients	Up and down		RMS-Illumina	[71]
28S-Cm4506	FBL depletion	Down	SNORD35A (RPL13A)	RMS-Illumina	[44]
	SNORA23 depletion/overexpression	Up/down	SNORD35B (RPS11)	RTL-P	[48]
	AES depletion	Down		RTL-P	[55]
	ß-thalassemic patients vs. healthy donor	Up		RTL-P	[69]
28S-Gm4593	FBL depletion	Down	SNORD78	RMS-Illumina	[44]
	ZFAS1 depletion/overexpression	Up/down	(GAS5)	RTL-P/DBPST	[47]
	Cell lines, between breast cancer patients	Up and down		RMS-Illumina	[71]
	DLBCL patients vs. healthy donor, brain, liver and skeletal muscle	Up in DLBCL, Up in brain and liver		RMS-Ion Torrent	[70]

^1^ Based on Biotoul database; ^2^ RMS = RiboMeth-Seq.

## 6. In Vivo Variability of rRNA 2′Ome in Model Organisms

In addition to revealing rRNA 2′Ome diversity in yeast, human cell lines, and samples, the development of RNA-seq-based approaches brought a significant insight into the mapping of rRNA 2′Ome sites at the entire organism/organ level in different plant and animal models, therefore enlarging the number of models available to elucidate the role of rRNA 2′Ome variability in living organisms (Figure 2).

### 6.1. Trypanosoma Brucei

In contrast to multicellular organisms, which have differentiated and specialized cells to perform a variety of functions and adapt to environmental conditions, *Trypanosoma brucei* is a unicellular eukaryotic parasite that cycles between two hosts (fly vs. affected animals including humans) and shows extremely coordinated adaptations to their surrounding environment [74]. Recently, a map of 99 highly confident 2′Ome sites was established on *T. brucei* rRNA using the combination of three different high-throughput technologies (RibOxi-seq, RiboMeth-seq and 2′-OMe-seq) [75]. These sites identified at the two life stages of *T. brucei*, including 31 2′Ome sites in the small subunit rRNA, 34 in the large subunit α, and 38 in the large subunit β.

In addition to the mapping of rRNA 2′Ome, the quantitative data generated by RiboMeth-seq allowed researchers to evaluate the intra- and inter-variability in *T. brucei*. First, this study showed that not all sites are equally methylated, but a subset of rRNA 2′Ome are partially methylated, as observed in cell lines and human samples. Thus, within a single sample, a mix of rRNA molecules methylated and non-methylated co-exist. Second, it appears that some sites display different levels of 2′Ome when comparing the two life cycles of *T. brucei*. Interestingly, most of these modulated 2′Ome sites are located around the functional domains of the ribosome (A, P, and peptidyl-transferase center sites). As in many cellular models and human samples, no correlation between the level of 2′Ome and SNORDs expression was established. Therefore, these results support the natural existence of different populations of ribosomes differentially methylated, that might support specific functions during *T. brucei* life cycles and contribute to the slight but recurrent adaptation of this living organism to differences in nutrients and temperature between the two hosts.

### 6.2. Arabidopsis Thaliana

Arabidopsis thaliana, the first plant to have its genome sequenced [76], is a widely used model organism in biochemistry, genetic development, and physiology. However, a complete cartography of rRNA 2′Ome sites was missing until recently. Up to now, the identification of rRNA 2′Ome sites in *A. Thaliana* has mainly consisted in predictions through sequence homology with SNORDs and all predicted sites could not be validated. Recently, two studies employed RiboMeth-seq for mapping rRNA 2′OMe sites in *A. thaliana*, thus providing a more confident cartography [77,78]. Hence, both studies identified 117 and 111 2′Ome sites, the difference potentially being due to either distinct statistical methods to identify new candidates or distinct biological materials, i.e., 3-week-old leaves plant seedlings [78] vs. ∼9 days light-grown seedlings [77]. Nevertheless, 110 2′Ome sites were found in common (2 sites in 5.8S, 35 in 18S, and 73 in 28S rRNAs), indicating that we are close to a complete mapping of methylated sites in *A. thaliana*.

Finally, these two studies also showed that the modulation of ribosome biogenesis factors affects rRNA 2′Ome in plants, as observed in cell lines [77,78]. Thus, the expression of Nucleolin (transcription and processing of pre-rRNA) and Fibrillarin mutants induced a general decrease in rRNA 2′Ome levels compared to wild-type plants. Moreover, it has to be noted that, in wild-type plants, only a subset of sites is partially methylated, indicating the existence of intra-variability in rRNA 2′Ome as well as inter-variability, although experimentally-driven, in *A. thaliana*.

### 6.3. Danio Rerio

*Danio rerio* (zebrafish) is another widely used model organism in biomedical research and developmental biology. Recent studies have shown that zebrafish expressed two distinct versions of 5S, 5.8S, 18S, and 28S rRNAs during early embryogenesis [79,80,81], where oocyte-specific rRNAs (i.e., maternal rRNAs) switch to somatic rRNA variants, the nucleotide sequences of which are different from maternal rRNAs. It has been hypothesized that these differences in rRNA sequences may have major functional consequences by affecting the ribosome:mRNA interactions and leading to specific translations of either maternal or somatic mRNAs. Interestingly, this shift from maternal to somatic rRNA molecules is also observable at the rRNA 2′Ome level [82].

First, a complete map of 98 2′Ome sites was established by compiling data issued from (i) RiboMeth-seq, (ii) primer extension assays and/or (iii) putative SNORD expression analyses, in pooled samples issued from four developmental stages (unfertilized eggs, the 32-cell, 12-somite, protruding mouth and tail) in 1-year old (adult) fish. Three 2′Ome sites were identified in 5.8S, 35 in 18S, and 60 in 28S rRNAs. Second, it appears that, although 97 sites were common between the two maternal and somatic rRNA types, one site was somatic-specific (28S-Cm3916). Moreover, a comparison of rRNAs issued from the maternal and somatic stages and adults revealed an increased methylation level of a subset of 2′Ome sites. In adult zebrafish tissues, the majority of 2′Ome sites seemed almost fully methylated and only few sites were partially methylated. As in other model organisms, natural variations of 2′Ome occurs in zebrafish, within particular developmental stages and adult tissues, therefore questioning the role of rRNA 2′Ome in development and tissue homeostasis.

### 6.4. Mus Musculus

Mouse remains an important model organism belonging the class of Mammalia and the super-order of Euarchontoglires, which contains both Rodentia and Primate. Therefore, mice share many developmental, biological, and genomic features with primates which makes it a perfect model for studying human biology and diseases, and for developing therapeutic strategies. Therefore, the perspective of studying the regulation and the function of rRNA 2′Ome and its potential use as a therapeutic target in humans required a deep understanding of the 2′Ome in mice. Despite the lack of 2′Ome studies in mice using biochemical approaches, such as mass spectrometry or primer extension assays, recent works have started the mapping of mouse rRNA 2′Ome through high-throughput methods. In contrast to methods such as RiboMeth-seq, which relies on the assumption that a site is modified to determine a methylation score, and therefore requires validation with gold-standard methods, the 2OMe-seq developed by Incarnato et al. has recently allowed researchers to uncover potential bona fide 2′Ome sites in mice without needing prior extensive biochemical studies [31]. Interestingly, comparisons of 2′Ome sites between human HeLa cells and mouse Embryonic Stem Cells (mESCs) indicated that 2′Ome sites are extremely conserved in mice and humans (92 out of 93 in common). Interestingly, 2OMe-seq identified 12 de novo 2′Ome sites in humans, 11 of which are conserved in mouse. Surprisingly, 28S-Gm4196, which is one of the most conserved 2′Ome site from *E. coli* to humans [24] was not found in mice in this study, in contrast to others (Durand S and Marcel V, unpublished data; [83]). Finally, although 2OMe-seq is not the most robust method to quantify 2′Ome levels, these data suggest nonetheless that the level of 2′Ome on specific sites is different between human and mouse. These species-specific 2′Ome cartographies suggest (i) that this diversity in location of 2′Ome sites in mouse ESCs is distinct from that of human HeLa cells; and (i) that mouse ESCs may possess ribosomes with compositional heterogeneity at the rRNA 2′Ome level.

Another milestone has been recently reached by the team of Cavaillé, in respect to the understanding of the 2′Ome biological diversity in mice [83]. By performing RiboMeth-seq analysis on adult and E16.6/E9.5 embryo tissues, they uncovered a fascinating dynamic of 2′Ome during mouse development. First, albeit mouse adult tissues display close-to-full methylation rates on the majority of rRNA sites, few positions exhibit partially methylated rates among the same tissues (intra-variability), while some other sites present differential 2′Ome levels when comparing distinct tissues (inter-variability). Thus, adult tissues present a low, albeit existing, degree of rRNA 2′Ome intra-variability, which is relatively conserved in all tested adult tissues (i.e., brain, liver, lung, heart, and kidneys). In contrast, many 2′Ome sites are considerably less methylated in developing tissues compared to their adult counterparts, therefore emphasizing a stunning developmental regulation of 2′Ome during mouse development. Notably, two 2′Ome sites, the 18S-Am576 and 28S-Gm4593, exhibited an opposite regulation, the latter one being associated with a reduction in 2′Ome level in adult tissues, most likely due to a developmental regulation of its related-SNORD78. Altogether, this work demonstrates the existence of a regulated 2′Ome pattern in adult mice and embryo tissues and sheds the light on the rRNA-related, developmentally-regulated ribosomal heterogeneity in mice.

## 7. Conclusions and Perspectives

The development of omic approaches (Table 1) dedicated to profiling rRNA 2′Ome in cell lines, fresh tissues, and human biopsies during the last 5 years has definitively demonstrated that rRNA 2′Ome varies in real-life depending on the physio-pathological context (Figure 2).

Intriguingly, not all methylated rRNA positions can tolerate a lack of 2′Ome. Indeed, it appears that only about one-third of methylated rRNA positions co-exists as a mix of methylated and non-methylated nucleotides. Due to the high diversity of technical, methodological, and statistical approaches used for profiling rRNA 2′Ome, it nevertheless remains difficult to firmly establish a genuine list of stable and variable sites, and further meta-analyses will therefore be required (Table 2). Similarly, we cannot rule out that unknown rRNA positions could be methylated in particular biological contexts. In addition, such techniques could not determine whether variations of rRNA 2′Ome reflect an intra-cellular 2′Ome heterogeneity (i.e., a mix of differentially methylated rRNA molecules within a single cell), or, in contrast, reveal a cell-homogenous 2′Ome that fluctuates between cell sub-populations. Finally, further studies should focus on establishing whether 2′Ome at variable positions co-exist or not on the same rRNA molecules or if they are, in contrast, mutually exclusive, therefore complexifying even further the degree of rRNA 2′Ome heterogeneity. As a matter of fact, the recent demonstration that single-molecule sequencing using nanopore technology can be used to map rRNA chemical modifications opens up novel perspectives in the field [84].

Altogether, these data clearly strengthen the hypothesis that rRNA 2′Ome may contribute to the establishment and/or maintenance of particular phenotypes, probably through a finely tuned control of translation, including in cancer. However, we cannot totally rule out that some variations of 2′Ome in vivo rather reflect a technical issue or biological noise and do not possess any functional roles. Few studies have ever addressed this question and many further in-depth studies are required to fully understand the role of rRNA 2′Ome in regulating particular phenotypes. One could hypothesize that, similar to observations made in *S. cerevisiae*, variable 2′O methylated positions may not be strictly essential for sustaining the bulk of cellular functions at a steady state, but could alternatively play a subtle role in particular conditions, such as, for instance, by adapting to rapid changes in environmental cues or responses to stress. Nevertheless, thanks to the recent advance in technologies dedicated to the rRNA epitranscriptomics, we now can firmly claim that rRNA 2′Ome varies naturally in real-life.

## Figures and Tables

**Figure 1 cells-10-01948-f001:**
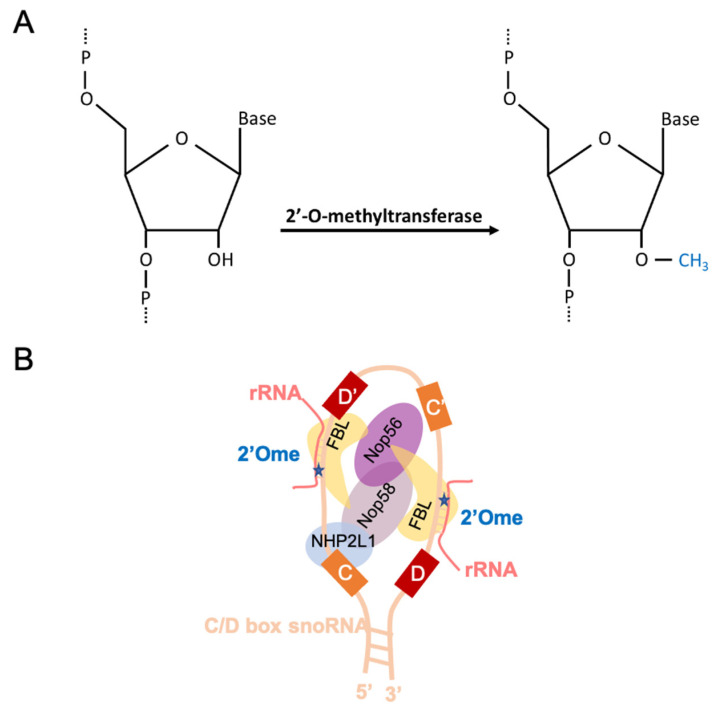
Schematic representation of a 2′-O-methylated ribose (**A**) and a C/D box snoRNP (also named the methylation complex) (**B**) The C/D box snoRNP includes the methyltransferase protein Fibrillarin (light yellow), NOP56 (purple), NOP58 (light brown), NHP2L1 (light blue), and the C/D snoRNA (light orange). Boxes C/C’ (orange) and D/D’ (red) are shown. The rRNA forms an RNA:RNA duplex with the C/D snoRNA. The FBL catalytic domain 2′O-methylates the fifth nucleotide downstream the boxes D or D’ and indicated by a blue star.

**Figure 2 cells-10-01948-f002:**
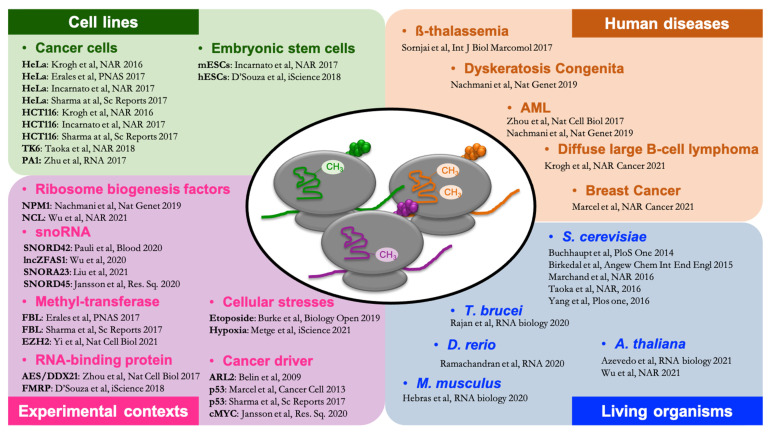
Variability of rRNA 2′Ome in living organisms, biological processes, and human diseases.

**Table 1 cells-10-01948-t001:** Description of the different methods used to quantify and detect the 2′Ome.

Method	Principle	2′Ome Site	2′Ome Detection	Reference
Primer Extension	The presence of a 2′Ome residue causes the Reverse Transcriptase to pause at low dNTP concentration, the pause being directly observable on a gel using a radioactive/fluorescent primer	site specific	qualitative	[28]
RTL-P	The presence of a 2′Ome residue decreases the reverse transcription efficiency at low dNTP concentration, the pause being quantified by regular or quantitative PCR	site specific	semi-quantitative	[29]
Mass Spectrometry	Detect rRNA fragments or nucleosides, the mass of which is shifted by specific modification including 2′Ome	global overview of all sites + site specific	quantitative (absolute)	[30]
RiboMeth-seq	The presence of a 2′Ome inhibits alkaline-mediated hydrolysis of the 3′-adjacent phosphodiester bond and is then detected by RNA-seq (Illumina or Ion Torrent sequencing)	global overview of known sites + site specific	quantitative (absolute)	[17]
2′OMe-seq	The presence of a 2′Ome residue causes the Reverse Transcriptase to pause at low dNTP concentration and is then detected by RNA-seq (Illumina)	global overview of all sites + site specific	quantitative (relative)	[31]
RibOxi-seq	The 3′-terminal ribose of a 2′Ome residue is resistant to periodate cleavage. After RNA fragmentation using the endonuclease Benzonase and periodate oxidation, RNA fragments with a 2′Ome group at their 3′end are enriched and detected by RNA-seq	global overview of all sites	qualitative	[32]

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
