# Peer review of "2′O-Ribose Methylation of Ribosomal RNAs: Natural Diversity in Living Organisms, Biological Processes, and Diseases"

_cells, 2021, doi:10.3390/cells10081948_

Round 1

Reviewer 1 Report

This review article is enjoyable to read and describes in an accurate and comprehensive manner our understanding of the function of rRNA 2'-O-ribose methylations. Specifically, the authors describe and discuss in a nuanced manner observations indicating that ribose methylation levels at some rRNA positions are variable, depending on the cell type, biological or even pathophysiological contexts. This manuscript is relevant because it is part of a growing research theme: the specialized ribosome hypothesis which promotes the idea that all ribosomes may not be functionally equivalent

I have a few very minor comments which are listed below.

On several occasions throughout the manuscript, the authors mention partially methylated sites by evoking a score lower than 0.9. I think it is necessary to be more precise (see also my comments on the table) and to give the reader an idea of the extent of the changes observed. In other words, are the observed changes very strong (score <0.2-0.3) or rather modest or even weak (score ~ 0.7-0.8)?

A somewhat provocative comment: can we rule out the trivial possibility that the variability observed at some rRNA sites simply results from technical issues and/or represents biological noise without any functional relevance in vivo? Since our understanding of the observed changes remains patchy, this possibility should be raised. In other words, "correlation does not imply causation".

Table 1 is very useful. However, I propose to improve it slightly by adding columns indicating: (i) the associated box C/D snoRNA; (ii) whether the change is down or up (simply by arrows) and (iii) the amplitude of these changes (RiboMeth scores or % of methylation when available)

Lane 54 -

“Accordingly, this vision was supported by observations using non-quantitative methods that, in the two historical models Saccharomyces cerevisiae and Xenopus laevis, rRNAs are fully methylated at almostall specific positions in all ribosomes and that the loss of single rRNA 2’Ome is lethal [6,9,10]”.

I'm not sure what the experimental evidence was at the time that suggested that loss of single rRNA 2’Ome is lethal. Can the authors clarify?

Lane 98 -

“Interactions between SNORDs and rRNAs depend on the particular structure of the methylation complex provided by NOP58, NOP56 and NHP2L1 (or 15.5kDa) in humans. Among archea and eukaryotes, some 2’Ome sites do not depend on the snoRNP methylation complex but instead on individual site-specific rRNA methyl-transferases [22]”

I suggest that the authors explicitly name in the text these few rare 2'-O-methylations whose synthesis does not depend on fibrillarin.

Lane 136 -

“First, the recent comparison of the 2’Ome patterns in chromatin-associated (i.e., transcribed) and mature rRNAs showed that the majority of the 2’Ome occurring on the….”

I recommend changing "transcribed" to "nascent transcripts"

Lane 144 -

“Conditional depletions of Nop1 alleles are lethal and result in both impairments of pre-rRNA 2’Ome and processing over long period of times”.

The authors should point out that one of the Nop1 alleles, if I remember correctly nop1.3, is deficient for 2'-O-methy transferase activity because it has a mutation in the SAM binding domain.

Lane 405-

 “Altogether, these studies shed light on the role of specific 2’Ome in normal hematologic functions and leukemogenesis”.

What are the strong experimental data that establish a causal link between rRNA methylation changes and biology of blood cells?

Lane 600 -

“In vivo variability of rRNA 2’Ome in metazoan model organisms”

I draw attention to the fact that Trypanosoma and A. thaliana are not Metazoan. They are Excavata and Archaeplastida, respectively. The title should therefore be changed, perhaps simply by indicating model organisms instead.

Author Response

We would like to thank the Reviewer 1 for his/her enthusiasm et his/her valuable comments and suggestions. Here is a point-by-point response to the Reviewer’s comments:

  • On several occasions throughout the manuscript, the authors mention partially methylated sites by evoking a score lower than 0.9. I think it is necessary to be more precise (see also my comments on the table) and to give the reader an idea of the extent of the changes observed. In other words, are the observed changes very strong (score <0.2-0.3) or rather modest or even weak (score ~ 0.7-0.8)?

When we mention “partially methylated with a c-score <0.9”, we refer to the intra-variability present within a single condition. When comparing two conditions, we then refer to inter-variability and thus compare ΔC-score. We think the reviewer is referring to this latter case.

We agree with the reviewer that providing a more precise insight into the 2’Ome inter- variability will be very valuable. However, we are confronted to a high diversity of approaches associated with different units, relative/absolute quantifications and used in many different studies, which cannot therefore be rigorously compared with each other’s. Second, even though the amplitude of some changes can be high in some studies, statistical analyses are not always provided and make appreciations of changes quite delicate. Hence, to not confuse the readers, we rather highlight studies and overall conclusions without emphasizing on C-score quantifications that could be misleading.

  • A somewhat provocative comment: can we rule out the trivial possibility that the variability observed at some rRNA sites simply results from technical issues and/or represents biological noise without any functional relevance in vivo? Since our understanding of the observed changes remains patchy, this possibility should be raised. In other words, "correlation does not imply causation".

We totally agree with the reviewer’s comment. Technical issues as well as biological noise could result in variation in rRNA 2’Ome levels. To address this crucial point, several times in the manuscript, we raised the lack of functional characterizations of 2’Ome changes on biological functions (lane 340, lane 743) so far. Moreover, we discussed the fact that variations in rRNA 2’Ome levels could simply result from the hyper-proliferation rate of cancer cells thus inducing a limiting rate effect of increased rRNA synthesis associated with no change in expression of the different components of the methylation complex (lines 602-608).

To emphasize this main point, we added a sentence to better highlight this hypothesis (line 1057-1059) of the revised manuscript).

  • Table 1 is very useful. However, I propose to improve it slightly by adding columns indicating: (i) the associated box C/D snoRNA; (ii) whether the change is down or up (simply by arrows) and (iii) the amplitude of these changes (RiboMeth scores or % of methylation when available)

We added column (i) and (ii) as suggested by the Reviewer. Regarding to (iii), we think that adding absolute value might be misleading due to usage of different units/methods (please refer to the answer provided in 1).

  • Lane 54 -“Accordingly, this vision was supported by observations using non-quantitative methods that, in the two historical models Saccharomyces cerevisiae and Xenopus laevis, rRNAs are fully methylated at almost all specific positions in all ribosomes and that the loss of single rRNA 2’Ome is lethal [6,9,10]”. I'm not sure what the experimental evidence was at the time that suggested that loss of single rRNA 2’Ome is lethal. Can the authors clarify?

We understand our statement could be confusing. We corrected the sentence accordingly.

  • Lane 98 -“Interactions between SNORDs and rRNAs depend on the particular structure of the methylation complex provided by NOP58, NOP56 and NHP2L1 (or 15.5kDa) in humans. Among archea and eukaryotes, some 2’Ome sites do not depend on the snoRNP methylation complex but instead on individual site-specific rRNA methyl-transferases [22]”. I suggest that the authors explicitly name in the text these few rare 2'-O-methylations whose synthesis does not depend on fibrillarin.

We added a sentence according to the reviewer’s suggestion (line 124-130). We provided the name of eukaryotic site-specific rRNA methyl-transferases as well as bacteria stand-alone rRNA methyl-transferases.

  • Lane 136 -“First, the recent comparison of the 2’Ome patterns in chromatin-associated (i.e., transcribed) and mature rRNAs showed that the majority of the 2’Ome occurring on the….”. I recommend changing "transcribed" to "nascent transcripts"

We corrected as requested by the reviewer.

  • Lane 144 -“Conditional depletions of Nop1 alleles are lethal and result in both impairments of pre-rRNA 2’Ome and processing over long period of times”. The authors should point out that one of the Nop1 alleles, if I remember correctly nop1.3, is deficient for 2'-O-methy transferase activity because it has a mutation in the SAM binding domain.

As far as we know, the reference we cited performed only NOP1 depletion using a conditional allele. The reviewer refers to Tollervey et al. Cell 1993 that we cite just above. We corrected to cite the reference correctly and added a sentence regarding the nop1.3 mutant (line 193-195).

  • Lane 405- “Altogether, these studies shed light on the role of specific 2’Ome in normal hematologic functions and leukemogenesis”. What are the strong experimental data that establish a causal link between rRNA methylation changes and biology of blood cells?

Nachmani et al. performed NPM1 depletions on the human erythroleukemia cell line and observed reduction of colony formation and increased differentiation upon hemin treatment. These effects could be phenocopied by knocking out individual snoRNA candidates (CRISPR). Moreover, these phenotypes caused by NPM1 depletion could be partially rescue by expression of single candidate snoRNAs and fully rescue by FBL over-expression. In addition, acute in vivo depletions of NPM1 in adult mouse hematopoietic stem cells cause features of bone marrow failure including dysmegakaryopoiesis, defective erythroid maturation, dysplastic and low platelet counts, dysplastic neutrophils and extensive hematopoietic precursor cells proliferation and apoptosis, although the role of 2’Ome defects in these phenotypes was not directly addressed. We though these observations were supporting the implication of 2’Ome in hematologic functions but we however understand the reviewer’s comment suggesting some direct pieces of evidence are still lacking. We modified the manuscript to nuance our conclusion.

  • Lane 600 - “In vivo variability of rRNA 2’Ome in metazoan model organisms”. I draw attention to the fact that Trypanosoma and A. thaliana are not Metazoan. They are Excavata and Archaeplastida, respectively. The title should therefore be changed, perhaps simply by indicating model organisms instead.

We thank the reviewer for pointing out this mistake. We corrected the manuscript accordingly.

Reviewer 2 Report

In the manuscript “2’O-ribose methylation of ribosomal RNAs: natural diversity in living organisms, biological processes, and diseases”, JAAFAR and colleagues extensively describe the diversity of 2’O-ribose methylation post-transcriptional modification in ribosomal RNAs (rRNAs). This review is up to date, very complete and based on an extensive analysis of the literature. It still suffers from minor drawbacks that should be fixed prior to publication.

  • The abstract is carelessly edited and quite confusing. Should be shortened to make it more accurate (eg “occur naturally in living organisms, various biological processes and diseases” does not make sense to me. The biological processes and diseases also occur in living organisms; ‘both within a single cell line”: both refers to what? and so on)

  • Page 2, line 49: to my knowledge pseudo-uridylation is a base modification

  • Page 3: a brief overview of rRNA 2’Ome

A figure displaying the modification is necessary, certainly showing the chemical reaction involved

Line 98: 15.5kDa refers to what??

  • Page 5:

A short chapter is necessary for bacteria, and before yeast, even if only three modifications were identified

  • Page 6, line 159, references 30, 31 do not describe slight impacts but rather “strong ones”

  • Lines 169-171

The decoding center should be better described – a figure would be welcome -

  • Page 8, from line 230

The following chapters are very long and rather descriptive. They would benefit from a couple of figures or even a sharp shortening

  • A box describing the new techniques that allowed the current data to be discovered would be welcome

  • Page 18, the table arrives too late

  • Nothing about structural studies (X-ray and cryo-EM) of the ribosome. Add a short paragraph describing the modifications observed with those techniques

  • An overall figure of the ribosome pointing out the important modifications or the crucial sites is necessary

Author Response

We would like to thank the Reviewer 2 for his/her enthusiasm et his/her valuable comments and suggestions. Here is a point-by-point response to the Reviewer’s comments:

in the manuscript “2’O-ribose methylation of ribosomal RNAs: natural diversity in living organisms, biological processes, and diseases”, JAAFAR and colleagues extensively describe the diversity of 2’O-ribose methylation post-transcriptional modification in ribosomal RNAs (rRNAs). This review is up to date, very complete and based on an extensive analysis of the literature. It still suffers from minor drawbacks that should be fixed prior to publication.

  • The abstract is carelessly edited and quite confusing. Should be shortened to make it more accurate (eg “occur naturally in living organisms, various biological processes and diseases” does not make sense to me. The biological processes and diseases also occur in living organisms; ‘both within a single cell line”: both refers to what? and so on)

We apologize if the abstract is confusing. We carefully edited the abstract to make it more concise and straightforward as suggested by the reviewer.

  • Page 2, line 49: to my knowledge pseudo-uridylation is a base modification

As pseudo-uridylation is a base isomerization, we agree it can be considered as a base modification. We initially wanted to separate from other base modifications such as methylation. We corrected the manuscript accordingly.

  • Page 3: a brief overview of rRNA 2’Ome. A figure displaying the modification is necessary, certainly showing the chemical reaction involved

We added a new figure (Figure 1) as suggested by the reviewer that illustrates the chemical 2’Ome modification and the composition of the rRNA 2’Ome complex.

  • Line 98: 15.5kDa refers to what??

“15.5kDa” refers to the alternative name for NHP2L1, commonly found in the literature. We edited the manuscript to clarify it.

  • Page 5: A short chapter is necessary for bacteria, and before yeast, even if only three modifications were identified.

It is our intention to focus our review on eukaryotes. Moreover, mentioning rRNA 2’Ome in procaryote will required an introduction on ribosome biogenesis as well as additional information regarding 2’Ome functions in procaryotes to feel exhaustive. Considering the complexity and the size of our review, as additionally raised by the reviewer 2, we prefer to not include any more material on bacteria but instead refer to excellent review covering this topic. However, when possible, we included few references to bacteria to emphasize the mechanistic conservation between organisms (line 134-137 for example).

  • Page 6, line 159, references 30, 31 do not describe slight impacts but rather “strong ones”

 We do not agree with the reviewer and maintain that “loss of individual modifications in yeast only impacts slightly, or not at all, cell viability or ribosome functions”. This conclusion is based on our own reading of the data discussed but also from the one of the authors as detailed in the abstracts of references we cited, such as for example:

Liang et al Mol Cell 2007 : “Blocking one to two modifications has no apparent effect on cell growth, whereas loss of three to five modifications impairs growth…”

Liang et al RNA 2009: “Loss of most modifications individually has no apparent effect on cell growth. However, deletions of 2-3 modifications…”

  • Lines 169-171 : The decoding center should be better described – a figure would be welcome

We added a sentence in the text to better describe the decoding center and the PTC (line 202-2014).

  • Page 8, from line 230 : The following chapters are very long and rather descriptive. They would benefit from a couple of figures or even a sharp shortening

We agree with the reviewer 2 that these chapters are very long and might feel descriptive sometimes.

However, these chapters are also covering the most abundant literature in the field, and we decided to provide the reader with a comprehensive review of this literature. We do not feel that a sharp shortening will be possible without removing essential information for the reader. As suggested by reviewer, a figure was already included (former Figure 1) and provides a very concise summary of studies described in these chapter.

  • A box describing the new techniques that allowed the current data to be discovered would be welcome

Excellent reviews have been dedicated to detail, compare and discuss the limits of each technic (Motorin and Marchand Genes 2018, Kroghs et al Methods 2019). Thus, to avoid redundancy with these reviews and as it could be informative to have a brief overview of these as suggested by the reviewer, we added a short table (Table 1 of the revised manuscript) to briefly described it. In addition, we provide a sentence indicating the reference to easily find these well-done reviews.

  • Page 18, the table arrives too late

We moved Table 1 after Chapter 5, as suggested by the reviewer.

  • Nothing about structural studies (X-ray and cryo-EM) of the ribosome. Add a short paragraph describing the modifications observed with those techniques

 We edited the manuscript to add these studies as suggested by the reviewer (line 326-330).

  • An overall figure of the ribosome pointing out the important modifications or the crucial sites is necessary

We think such a figure will be redundant with Table 2 (revised manuscript), which already mentions the most variable sites as well as some additional information (context, techniques as well as corresponding snoRNAs/host genes and variation as requested by other reviewers).

We also tried to point out important sites on the ribosome structure (see the supplemental figure attached). However, we do think that this figure is too confusing and not enough informative. Therefore, we did not include it in the revised manuscript.
